# Comprehensive Investigation of Cooling, Heating, and Power Generation Performance in Adsorption Systems Using Compound Adsorbents: Experimental and Computational Analysis

**Zisheng Lu**

Chongqing Research Institute, Institute of Refrigeration and Cryogenics, Shanghai Jiao Tong University, Dongchuan Rd. 800#, Shanghai 200240, China; zslu@sjtu.edu.cn; Tel.: +86-21-3420-6309

**Abstract:** The extensive utilization of petrochemical energy sources has led to greenhouse gas emissions, the greenhouse effect, the frequent occurrence of extreme weather events, and the severe degradation of Earth's ecosystems. The development of renewable energy technologies has become an inevitable trend. This paper investigates an adsorption-based cooling/heating/power generation technology driven by low-grade solar thermal energy. The research results demonstrate that the adsorption performance of vermiculite compound adsorbents impregnated with LiCl solution is superior to those impregnated with $CaCl_2$ solution, with the former exhibiting adsorption at lower p/po partial pressure ratios. Furthermore, at an adsorption bed temperature of 25 °C and a p/po partial pressure of 0.8, the adsorption cooling performance of Comp. 2 compound adsorbent impregnated with LiCl solution reaches 5760.7 kJ/kg, with a coefficient of performance (COP) of 0.75, heating performance of 9920.8 kJ/kg, COPh of 1.51, and power generation capacity of 10.6 kJ/kg. This research contributes to the advancement of sustainable energy technologies and the mitigation of environmental impacts associated with petrochemical energy sources.

**Keywords:** compound adsorbents; water; cooling; heating; power generation





## 1. Introduction

The pressing need to discover sustainable and renewable energy sources has arisen due to the stark contrast between the rapid surge in energy demand and the dwindling reserves of conventional fossil fuels. Renewable energy and residual heat emerge as highly appealing alternatives to conventional fossil fuels, offering a sustainable and ecologically friendly energy reservoir that holds the potential to curtail greenhouse gas emissions and alleviate the repercussions of climate change. Unlike fossil fuels, which are finite and progressively challenging to extract, renewable energy sources such as solar and wind are abundant and widely accessible. Simultaneously, residual heat stands as an accessible resource within numerous industrial processes, which can be effectively harnessed to generate electricity or supply heating services. These resources, encompassing renewable energy sources and residual heat, have the capacity to generate electricity, cooling, or heating with minimal to negligible greenhouse gas emissions, thereby diminishing air pollution and attenuating the adverse effects of climate change [1,2].

Numerous researchers have delved into the investigation of advanced adsorbents, notably metal–organic frameworks (MOFs), and their applications in various domains, including adsorption-based cooling systems. For instance, in the work of M. Rezk et al., a dedicated effort is made to enhance adsorption systems for cooling and desalination by leveraging advanced materials and innovative adsorbent bed configurations. Their study employs computational methodologies to assess the performance of a heat exchanger incorporating a copper-foamed adsorbent bed packed with MOF-801 adsorbent, juxtaposed against a baseline adsorbent comprising silica gel. Employing a multi-objective

optimization approach, the research aims to attain the optimal balance between coefficient of performance, specific cooling power, and clean water productivity. The outcomes unequivocally demonstrate that the MOF-801-based system surpasses its silica gel counterpart in terms of clean water production and specific cooling power. However, it's noteworthy that the latter exhibits superior cooling capacity and coefficient of performance, primarily attributable to its higher packing density [3].

In a similar vein, the research undertaken by S. Chumnanwat et al., explores the application of an innovative adsorbent coating technique on aluminum fins within an adsorption-based heat pump or chiller system. The methodology involves immersing an aluminum substrate into a solution containing the adsorbent, resulting in the formation of a compound adsorbent layer on the surface. The study rigorously evaluates the adhesion properties of this layer through a peeling test. Notably, the investigation identifies specific anodization conditions and the utilization of zeolite AQSOA-Z01 as the adsorbent, resulting in the formation of a thin aluminum oxide film layer with commendable adhesion strength and specific cooling capacity, making it a promising option for adsorption heat pump and chiller applications [4].

H. Banda's research delves into the exploration of graphene oxide's applicability as an adsorbent material within adsorption systems, drawing a comparative analysis against the performance of silica gel. The findings unequivocally underscore the considerable enhancements realized with graphene oxide, with a remarkable 44% boost in thermal efficiency, up to 57% improvement in adsorption capabilities, as well as notable advancements in specific daily water production, specific cooling power, coefficient of performance, and energy efficiency when juxtaposed with silica gel [5].

In the realm of experimental investigations, a hybrid adsorption system, strategically coupling a pretreatment module tailored for heavy metal removal with a unit focused on desalination and cooling, underwent scrutiny. This multifaceted system harnessed a trio of adsorbents, namely activated carbon, zeolite, and aluminum fumarate MOF. The outcomes attest to the system's impressive efficacy in the removal of heavy metals, yielding a prolific daily production of distilled water exceeding 260 L and generating a robust 6.9 kW of cooling power while boasting a commendable coefficient of performance of 0.26. These findings collectively emphasize the system's potential as a promising solution, not only for water purification but also for the eco-friendly production of clean cooling resources [6].

To elevate the adsorption efficacy of adsorbents, extensive scholarly investigations have delved into the realms of adsorption performance, heat and mass transfer characteristics, and the optimization of efficient sorption processes, particularly with compound adsorbents comprising physical adsorbents such as MOFs and hygroscopic salts. For instance, H. Zhao embarks on an exploration of MIL-100(Fe)'s water vapor adsorption properties within the context of sorption-based atmospheric water harvesting (SAWH). This endeavor involves a comparative analysis between methods involving solvents and those devoid of solvents for the preparation of MIL-100(Fe). The findings emanating from this study unequivocally indicate that MIL-100(Fe) synthesized in a solvent-free manner, alongside its compound adsorbent counterpart, $MgCl_2$@MIL-100(Fe), exhibits notable enhancements in adsorption performance and desorption characteristics. These improvements manifest through higher equilibrium adsorption capacities when contrasted with the conventional solvent-based synthesis approach. These insightful results not only underscore the potential application of solvent-free MIL-100(Fe) and its associated compound adsorbent within the realm of SAWH systems but also provide a compelling avenue for future research and technological advancement [7].

Throughout their investigation, the research team unearthed pivotal revelations. Specifically, they ascertained that MOF-801 displayed significantly superior adsorption (desorption) capacities in comparison to both silica gel and the 13X molecular sieve. The enhancements amounted to a substantial increase, respectively. In parallel, the scholarly endeavors led by Bo Han and his collaborators have been dedicated to a multifaceted exploration of adsorption-based thermal energy storage [8]. This domain holds vast potential

as a foundational cornerstone for diverse heat-related processes encompassing cooling, heat pumps, desalination, power generation, water harvesting, and dehumidification. Their comprehensive investigation yielded remarkable findings: targeted modifications to the original MOFs, particularly the MIL-53 (Al) MOFs, resulted in alterations in hydrophilicity and hydrophobicity, leading to an enhancement in water loadings of up to 0.9 g/g. This augmentation was further exemplified through the accelerated water transfer facilitated by functionalized and protonated MIL-53 (Al) MOFs, demonstrating enhanced kinetics when compared to their unmodified counterparts. Notably, the ligand-extended MIL-53 (Al) MOFs showcased impressive capabilities, particularly in transferring water between environments characterized by high humidity (80% to 90% relative humidity) and those undergoing regeneration (30% relative humidity). This innovative approach yielded a promising thermal energy storage density (TESD) reaching up to 1.54 MJ/L. Sandra Jose and her colleagues undertook a comprehensive review focused on energy storage applications, with particular emphasis on compounds comprised of conducting polymers and metal-organic frameworks (CP/MOF). The findings underscore the versatility of CP/MOF compounds, indicating their potential extension into a range of domains, including electrochemical sensors, water splitting, and carbon dioxide reduction [9]. In a distinctive investigation, Fan Luo and colleagues delved into an innovative monolithic adsorbent derived from bi-metallic MOFs, designed specifically for solar-triggered atmospheric water harvesting. This bimetallic MOF exhibited an impressive specific surface area of 1203 $m^2 \cdot g^{-1}$ and a substantial pore volume measuring 0.51 $cm^3 \cdot g^{-1}$. Their efforts culminated in the successful fabrication of an advanced water harvesting system, ultimately showcasing a remarkable daily water yield of up to 1.9 g/g through the purpose-built monolithic adsorbent [10]. Majdi Amin's research revolved around the scrutiny of sorption-ejector systems, emphasizing the imperative need for further experimental validation of theoretical data. Within integrated sorption-ejector configurations, notable enhancements came to the fore: a noteworthy 9.8% reduction in power consumption, a 13.6% decrease in cooling capacity, and a substantial 8–60% improvement in the coefficient of performance compared to standalone sorption systems. Moreover, the overall coefficient of performance (COP) for combined adsorption-ejector systems exhibited a significant increase, registering increments of 0.33 and 1.47, respectively [11]. Turning attention towards energy storage, Jingwei Chao and their team harnessed zeolite/$MgCl_2$ compound sorbents with the aim of augmenting energy storage density. Their outcomes revealed impressive average energy densities, notably peaking at 686.86 kJ/kg for heat storage and 597.13 kJ/kg for cold storage [12]. In another exploration led by Suboohi Shervani and their associates, the focus was on harnessing the potential of vermiculite/LiCl compound adsorbents for thermal energy storage. The results underscored a substantial enhancement in energy storage performance facilitated by this compound adsorbent. Significantly, the energy storage density could reach an impressive 159 kWh/$m^3$ under specific regeneration conditions, particularly at a temperature of 120 °C [13]. Finally, Faraz Ege and other research teams embarked on an assessment of MIL-101(Cr)-coated microchannels as adsorbents to amplify adsorption performance. The experimental findings vividly demonstrated the superior performance of these meticulously characterized coated channels across various domains, encompassing adsorption as well as adsorption-driven heating and cooling capabilities [14–18]. Efficient adsorbents are applied in adsorption systems for seawater desalination and adsorption refrigeration. For instance, activated carbon and zeolite are used as adsorbents, and MOF is coated on the adsorbents. Research results demonstrate a desalination capacity of 260 L/day and a refrigeration power of 6.9 kW [6]. In order to enhance the overall performance coefficient of the adsorption system, researchers utilized oxidized graphene as the adsorption material for adsorption refrigeration and seawater desalination. Research findings reveal that, compared to the silica gel system, the seawater desalination capacity of the oxidized graphene system increased by 44.4%, and the refrigeration performance improved by 29.5% [5]. The researchers employed an aluminum fumarate metal-organic framework, or SAPO-34/CPO-27(Ni) as the adsorption material for the study of adsorption-

based seawater desalination and adsorption refrigeration. Research findings demonstrate a significant enhancement in the coefficient of performance of the system [19,20].

As mentioned previously, numerous scholars have actively conducted research in the domains of adsorbents, adsorption cooling, and adsorption energy storage. Nevertheless, there has been a conspicuous dearth of comprehensive investigations into the innovative adsorption system capable of concurrently facilitating cooling, heating, and power generation through the utilization of compound adsorbents. The primary objectives of this research paper encompass the following aspects: Firstly, the development of a series of compound adsorbents is undertaken, accompanied by a comparative analysis of their structural characteristics and adsorption performance. Secondly, a thorough examination is conducted to assess their cooling capabilities, heating performance, and power generation efficiency.

The innovative aspects of this paper are primarily reflected in the following areas: (1) Novel High-Efficiency Composite Adsorbents: This study investigates high-porosity zeolites and immerses them in hygroscopic salt solutions such as calcium chloride and lithium chloride to create innovative composite adsorbents. Their high porosity enables the storage of a greater quantity of adsorbate solution. Calcium chloride enhances adsorption performance while considering the cost-effectiveness of composite adsorbent production. Lithium chloride effectively lowers desorption temperatures and improves adsorption capabilities. (2) Innovative Composite Cycle: Due to the enhanced adsorption performance of the novel composite adsorbents, this paper presents the possibility of constructing a composite cycle. This composite cycle can facilitate adsorption-based refrigeration, adsorption-based heating, adsorption-based desalination, and adsorption-based power generation. This approach can significantly enhance the overall cycle efficiency. (3) Expanded Application Scenarios: The system can be powered by both solar energy and industrial waste heat, making it suitable for a variety of applications, including island environments, deserts, and industrial settings. This versatility contributes to economic and environmental benefits.

## 2. Matrix-Salt Adsorbents and Test Setup

This paper outlines the fabrication process of four distinct matrix-salt adsorbents. These adsorbents feature porous media composed of activated carbon, vermiculite, and activated carbon/silica diatomite. To prepare them, the porous media were submerged in salt solutions with precise concentrations for a 12-h period. For comparison, two different salt solutions were utilized, and their respective parameters can be found in Table 1. Following immersion, the adsorbents underwent further treatment in an oven at 130 °C for 12 h, culminating in the formation of the matrix-salt adsorbents, as illustrated in Figure 1.

**Table 1.** Main components of matrix-salt adsorbents and parameters of immersion salt solutions.

| Type | Matrix | Salt Solution Components | | | |
|------|--------|-----------|-------|----------|---------|
| | | $CaCl_2$% | LiCl% | $MgCl_2$% | MIL101% |
| Matrix-salt 1 | Activated carbon | 10.9 | 14.8 | 0 | 1.8 |
| Matrix-salt 2 | Vermiculite | 10.9 | 14.8 | 0 | 1.8 |
| Matrix-salt 3 | AC/Diatomite | 10.9 | 14.8 | 0 | 1.8 |
| Matrix-salt 4 | Vermiculite | 30% | 0 | 0 | 0 |

The examination of the matrix-salt adsorbents was conducted employing a confocal microscope. When scrutinizing these materials, distinct characteristics emerged. Activated carbon displayed a relatively consistent pore size distribution, while activated carbon/silica diatomite exhibited a more varied pore structure, encompassing fine micropores to larger macropores. In contrast, vermiculite showcased a markedly dissimilar morphology compared to porous substances like activated carbon. Vermiculite assumed a "worm-like" configuration and featured a stratified structure characterized by a relatively substantial pore volume.

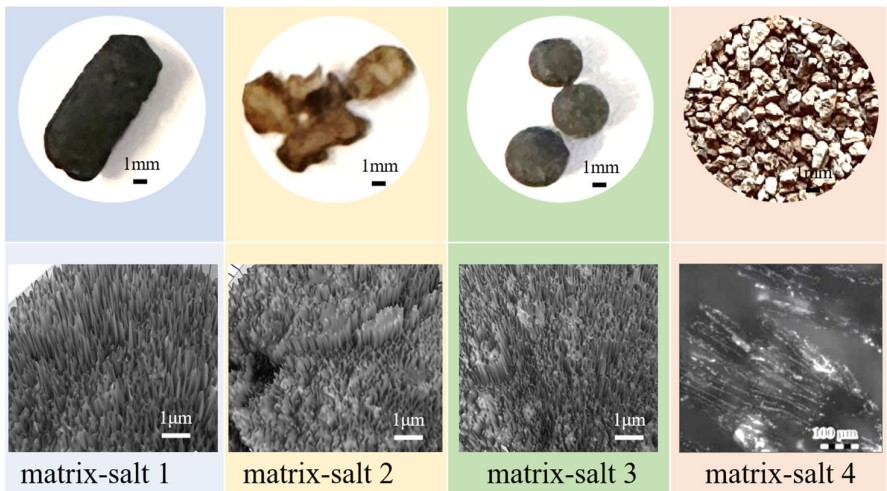

**Figure 1.** Photograph and confocal microscopy images of matrix-salt adsorbent.

To assess the adsorption capacity of the matrix-salt adsorbents, we employed an ASAP 2020 (Norcross, GA, USA) device, as depicted in Figure 2. The ASAP 2020 serves as a high-performance adsorption analyzer, facilitating the measurement of surface area, pore size, pore volume, and adsorption capacity of the matrix-salt adsorbents. Additionally, the pore structure of these matrix-salts was thoroughly analyzed utilizing a confocal microscopy device known as Smartproof 5 (White Plains, NY, USA). Smartproof 5 harnesses confocal imaging principles to achieve a remarkable 1.4-fold enhancement in traditional optical resolution. It boasts XY direction line resolution of up to 120 nm and a minimum Z-axis step accuracy of 1 nm.

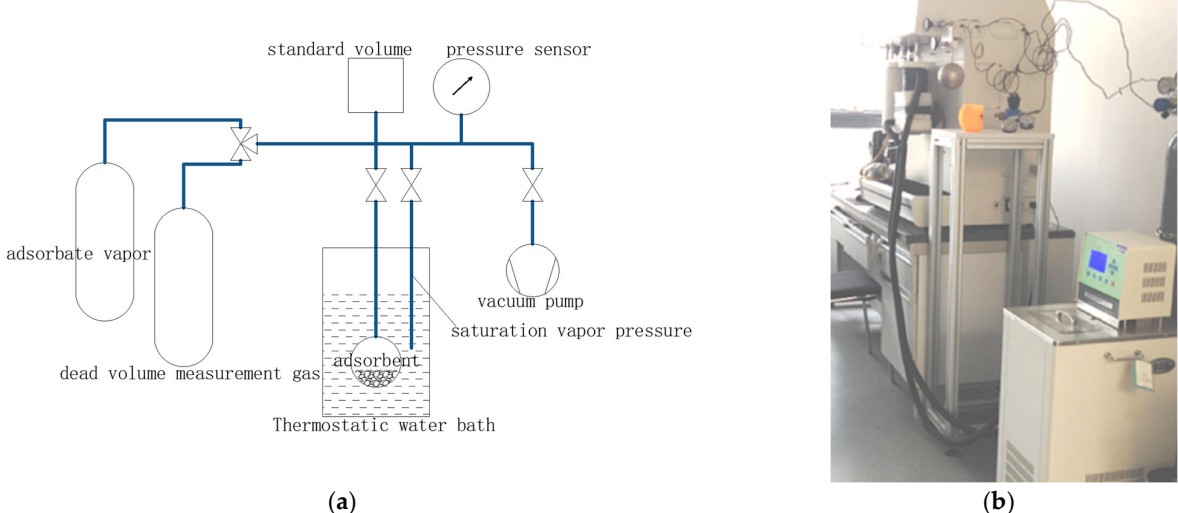

**Figure 2.** Experimental instrument: ASAP adsorption analyzer; (**a**) block diagram; (**b**) experimental setup.

For assessing adsorption performance and overall system performance, the following equations can be employed:

$$x = x_0 e^{-k[\ln \frac{p_{(Ts)}}{p_{(Ta)}}]^n} \tag{1}$$

where $x$ is the adsorption capacity, kg/kg; $x_0$ is the maximum adsorption capacity, kg/kg; $k$ is a constant; $p_{(Ts)}$ is the pressure at saturation temperature, Pa; $p_{(Ts)}$ is the pressure at adsorption temperature, Pa; and $n$ is a constant.

The equation for evaporation heat ($Q_{evaporat}$) is given by:

$$Q_{evaporat} = m_{adsorben}\left(x_{adsorptio.,end} - x_{adsorptio.,begin}\right) \cdot \left[h_{vaporizatio} + C_{p,wate}\left(T_{condensatio} - T_{evaporatio}\right)\right] \qquad (2)$$

where $Q_{evaporat}$ is the cooling capacity, kJ; $m_{adsorben}$ is the adsorbent mass, kg; $x_{adsorptio.,end}$ is the adsorption uptake at the end of the cycle, kg/kg; $x_{adsorptio.,begin}$ is the adsorption uptake at the beginning of the cycle, kg/kg; $h_{vaporizatio}$ is the latent heat of vaporization, kJ/kg; $C_{p,wate}$ is the specific heat capacity, kJ/(kg.K); $T_{condensatio}$ is the condensation temperature, K; and $T_{evaporatio}$ is the evaporation temperature, K.

The adsorption cooling capacity ($Q_{cooling}$) equation is given by:

$$Q_{cooling} = m_{adsorbed}C_p\left(T_{condensation} - T_{evaporation}\right) + m_{adsorbed}h_{vapor} \qquad (3)$$

where $Q_{cooling}$ is the adsorption cooling capacity, kJ/kg; $m_{adsorbed}$ is the mass of adsorbate adsorbed, kg/kg; $C_p$ is the specific heat capacity of the adsorbate, kJ/(kg.°C); $T_{condensation}$ is the condensation temperature, °C; $T_{evaporation}$ is the evaporation temperature, °C; and $h_{vapor}$ is the latent heat of vaporization, kJ/kg.

The equation for heating ($Q_{heating}$) is given by:

$$Q_{heating} = m_{bed}C_{p,bed}\left(T_{desorption} - T_{adsorption}\right) + \int_{T_{desorption,\ start}}^{T_{desorption,end}} m_{adsorbed}\ C_{p,adsorbate}dT + m_{adsorbed}h_{desorption} \qquad (4)$$

where $Q_{heating}$ is the heating capacity, kJ/kg; $m_{bed}$ is the mass of the bed, kg; $C_{p,bed}$ is the specific heat capacity of the adsorbate, kJ/(kg.°C); $T_{desorption}$ is the desorption temperature, °C; $T_{adsorption}$ is the adsorption temperature, °C; $C_{p,adsorbate}$ is the specific heat capacity of the adsorbate, kJ/(kg.°C); $m_{adsorbed}$ is the mass of adsorbate adsorbed, kg/kg; and $h_{desorption}$ is the heat of desorption, kJ/kg.

The equation for the cooling power of the cooling water ($Q_{cool,water}$) is given by:

$$Q_{cool,water} = m_{bed}C_{p,bed}\left(T_{desorption} - T_{adsorption}\right) + \int_{T_{adsorption,\ start}}^{T_{adsorption,end}} m_{adsorbed}\ C_{p,adsorbate}dT + m_{adsorbed}h_{desorption} \qquad (5)$$

where $Q_{cool,water}$ is the cooling power of the cooling water, kJ/kg; $m_{bed}$ is the mass of the bed, kg; $C_{p,bed}$ is the specific heat capacity of the adsorbate, kJ/(kg.°C); $T_{evaporation}$ is the evaporation temperature, °C; $T_{adsorption}$ is the adsorption temperature, °C; $C_{p,adsorbate}$ is the specific heat capacity of the adsorbate, kJ/(kg.°C); $m_{adsorbed}$ is the mass of adsorbate adsorbed, kg/kg; and $h_{desorption}$ is the heat of desorption, kJ/kg.

The coefficient of performance (COP) equation for an adsorption cooling system is defined as:

$$COP = \frac{Q_{cooling}}{Q_{heating}} \qquad (6)$$

where *COP* is the coefficient of performance; $Q_{cooling}$ is the cooling capacity, kJ/kg; and $Q_{heating}$ is the heating capacity, kJ/kg.

The condensation capacity ($Q_{condensation}$) equation can be expressed as:

$$Q_{condensation} = m_{cndensed}C_p\left(T_{desorption} - T_{condensation}\right) + m_{cndensed}h_{condensation} \qquad (7)$$

where $Q_{condensation}$ is the condensation capacity, kJ/kg; $m_{cndensed}$ is the mass of condensate water, kg; $C_p$ is the specific heat capacity of the condensate, kJ/kg; $T_{desorption}$ is the desorption temperature, °C; $T_{condensation}$ is the condensation temperature, °C; and $h_{condensation}$ is the heat of condensation, kJ/kg.

The coefficient of performance for heating ($COP_h$) equation in an adsorption heating system is given by:

$$COP_h = \frac{Q_{condensation} + Q_{cool,water}}{Q_{heating}} \qquad (8)$$

where $COP_h$ is the coefficient of performance for heating.



The power generation ($COP_h$) equation is given by:

$$w_{electricity} = c_{vapor,water} \cdot m_{vapor,\ water} \cdot \left( T_{desorption} - T_{condensation} \right) \cdot \mu_{expender} \cdot \mu_{generator} \qquad (9)$$

where $w_{electricity}$ is the power generation capacity, kJ/kg; $c_{vapor,water}$ is the specific heat capacity of the adsorbate, kJ/(kg.°C); $m_{vapor,water}$ is the mass flow of the water vapor, kg/s; $T_{desorption}$ is the desorption temperature, °C; $T_{condensation}$ is the condensation temperature, °C; $\mu_{expender}$ is the efficiency of the expender; and $\mu_{generator}$ is the efficiency of the power generator.

The advantages of matrix-salt adsorbents are shown as follows: Matrix-salt adsorbents, also known as hybrid adsorbents, offer a multitude of advantages in various applications, ranging from environmental remediation to industrial processes. These innovative materials combine different adsorbent components to create synergistic effects that enhance their performance. Below, we explore the numerous benefits associated with matrix-salt adsorbents. Enhanced Adsorption Capacity: Matrix-salt adsorbents often exhibit higher adsorption capacities compared to single-component adsorbents. This increased capacity results from the combination of multiple adsorbent materials, each with its own unique adsorption characteristics. Improved Selectivity: Tailoring matrix-salt adsorbents by carefully selecting their components allows for enhanced selectivity. This feature is particularly valuable in separating target matrix-salts from complex mixtures. Versatility: Matrix-salt adsorbents can be designed to target specific pollutants or contaminants, making them versatile tools for various applications, such as wastewater treatment, air purification, and gas separation. Regenerability: Many matrix-salt adsorbents are readily regenerable, allowing for multiple usage cycles. This property reduces operational costs and environmental impacts compared to one-time-use adsorbents. Faster Adsorption Kinetics: The synergistic effects of matrix-salt adsorbents can result in faster adsorption kinetics, enabling more efficient pollutant removal or gas separation processes. Improved Stability: The combination of different adsorbent materials often enhances the stability and durability of the matrix-salt adsorbents, ensuring a longer service life. Cost-Effectiveness: Matrix-salt adsorbents can be designed to use readily available and cost-effective materials, making them an economically viable option for various industries. Tailored for Specific Applications: These adsorbents can be tailored to suit specific application requirements, allowing engineers and researchers to design solutions that meet their unique needs. Reduced Environmental Impact: Matrix-salt adsorbents can help reduce the environmental impact of various processes by effectively removing harmful substances and minimizing waste generation. Innovation and Research Opportunities: The development of matrix-salt adsorbents continues to drive innovation in materials, science, and engineering, offering researchers exciting opportunities to explore new combinations and applications. Scalability: Matrix-salt adsorbents can be designed for scalability, making them suitable for both small-scale and large-scale operations. Compatibility with Existing Systems: In many cases, matrix-salt adsorbents can be integrated into existing treatment or separation systems with minimal modifications, providing a seamless upgrade option. Wide Range of Applications: Matrix-salt adsorbents find applications in diverse fields, including water purification, gas separation, pharmaceutical manufacturing, and more, making them indispensable in modern industries. Matrix-salt adsorbents represent a promising and versatile class of materials that offer numerous advantages across a wide spectrum of applications. Their ability to combine the strengths of different adsorbent components makes them a vital tool for addressing environmental and industrial challenges while promoting sustainability and cost-effectiveness. Continued research and development in this field promise to unlock even more potential benefits in the future.

The model and specification of the equipment are shown in Table 2. The schematic diagram of the adsorption cooling/heating/electricity generation system is depicted in Figure 3. The main working principle of the experimental system is based on the adsorbent's adsorption or desorption of the adsorbate. The adsorbent can be heated by heat

sources at different temperatures or cooled by cooling water at different temperatures. The adsorbate can evaporate at different temperatures or be condensed into a liquid. Under different operating conditions, the adsorption capacity of the adsorbent for the adsorbate varies. From the diagram, it is evident that it mainly consists of two constant-temperature water baths, an adsorption bed, an evaporative condenser, a vacuum pump, a pressure storage tank, a power generation device, and more. Its operating principle is as follows: Firstly, during the adsorption bed heating and desorption process, hot water from constant-temperature water bath 1 heats the adsorption bed, causing the matrix-salt adsorbent within the bed to desorb water vapor. Secondly, in the condensation process, water vapor is condensed into pure water within the evaporative condenser, and the condensation heat is carried away by constant-temperature water bath 2. Thirdly, during the cooling and adsorption process, cold water from constant-temperature water bath 1 cools the adsorption bed, and the adsorbent within the bed adsorbs the vapor evaporated from the evaporative condenser. At the same time, the adsorption bed releases adsorption heat, which is carried away by constant-temperature water bath 1. Fourthly, in the evaporative cooling process, the adsorbate water evaporates in the evaporative condenser, simultaneously generating cooling capacity, which is carried away by constant-temperature water bath 2. Fifthly, in the vortex power generation process, the high-pressure water vapor resulting from adsorption bed desorption is stored in the pressure storage tank, and this steam can drive a vortex generator for electricity generation. The experimental setup is illustrated in Figure 4.

**Table 2.** Model and specification of the equipment.

| No. | Name | Model | Specification | Accuracy |
|-----|------|-------|---------------|----------|
| 1 | Temperature sensor (Garland, TX, USA) | OMEGATJ36 K type | 0–1200 °C | 0.4% |
| 2 | Pressure sensor (Mainz, Germany) | WIKA S-20 | 0–6 MPa | 0.12% |
| 3 | Mass flowmeter (Monterey, CA, USA) | LONTROL DN15 | 0.6–3.6 m$^3$/h | 0.5% |

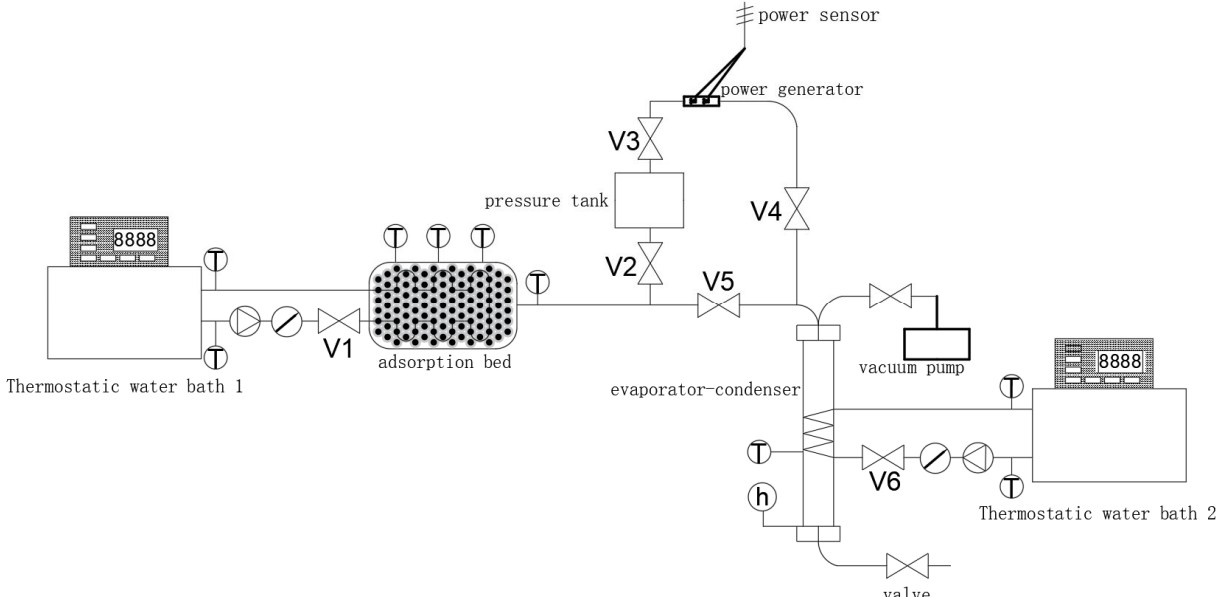

**Figure 3.** The adsorption cooling/heating/power generation system.

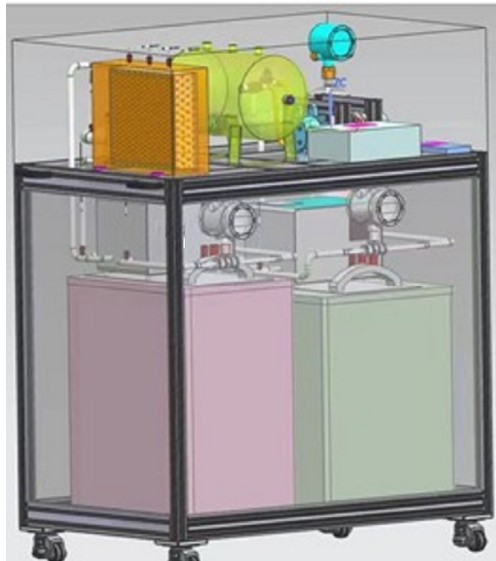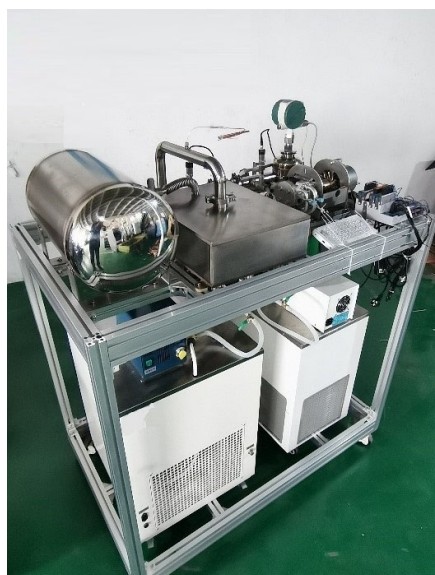

**Figure 4.** The setup of the adsorption cooling/heating/power generation system.

## 3. Results of Performance of Cooling, Heating and Power Generation

### 3.1. Performance of Cooling

The outcomes of adsorption performance assessments for the matrix-salt adsorbents are visually depicted in Figure 5. Upon a careful examination of the graph, it becomes apparent that the adsorption performance of matrix-salt adsorbents employing vermiculite as the porous medium demonstrates a relatively favorable performance and follows a comparable adsorption trend. It is essential to note, however, that the adsorption performance of vermiculite-based matrix-salt adsorbents impregnated with $CaCl_2$ exhibits a slight lag in comparison to those impregnated with LiCl and other adsorbents. This observed difference can primarily be attributed to the lower moisture sorption capacity of $CaCl_2$ when contrasted with LiCl. Furthermore, the graph reveals that at lower p/po (relative pressure) values, matrix-salt 2 exhibits a higher adsorption capacity when juxtaposed with matrix-salt 4. Additionally, matrix-salt 4 showcases a more fragmented adsorption curve in contrast to matrix-salt 2. This distinction can be ascribed to the broader spectrum of hydrates formed by $CaCl_2$ as opposed to those generated by LiCl. Figure 5 provides a visual representation of the adsorption performance of various matrix-salt adsorbents. Notably, vermiculite-based adsorbents demonstrate favorable performance trends, albeit with variations attributed to the choice of impregnating agent. Additionally, the differences in adsorption capacity and curve characteristics between matrix-salt 2 and matrix-salt 4 are elucidated, shedding light on the influence of the impregnation agent's properties on adsorption behavior. These findings contribute valuable insights to the understanding of matrix-salt adsorbents' performance in different configurations.

Figure 6 provides a graphical depiction of the cooling performance of the matrix-salt adsorbents, while Figure 7 furnishes valuable insights into the coefficient of performance (COP). Within the presented graphical data, a discernible pattern emerges in which the adsorption cooling performance exhibits an ascending trajectory with increasing evaporator pressure. When the focus is narrowed to matrix-salt adsorbents featuring vermiculite as the uniform porous substrate, a clear trend materializes wherein the COP tends to converge as the partial pressure (p/po) ascends. What is particularly noteworthy is the conspicuous prominence of matrix-salt 2, which demonstrates the most exceptional cooling performance among the evaluated matrix-salts. In stark contrast, matrix-salt 3 exhibits the least effective cooling performance within the defined parameters. To illustrate, at an adsorption bed temperature of 25 °C and a p/po partial pressure of 0.8, matrix-salt 2 boasts a remarkable cooling performance of 5760.7 kJ/kg, accompanied by a COP of 0.75. This is because

matrix-salt 2 has a high porosity of zeolite, and additionally, $CaCl_2$ and LiCl exhibit high hygroscopic properties. Therefore, matrix-salt 2 possesses superior adsorption capability and adsorption refrigeration performance. Conversely, under identical conditions, matrix-salt 3 manages to achieve only a modest cooling performance of 597.1 kJ/kg, coupled with a significantly lower COP of 0.46. Figure 6 visually encapsulates the cooling performance of the various matrix-salt adsorbents, revealing an upward trajectory in cooling performance as evaporator pressure increases. Moreover, a detailed examination of Figure 7 underscores the noteworthy disparities in COP among the matrix-salts, with matrix-salt 2 emerging as the frontrunner in cooling performance, while matrix-salt 3 lags behind under the specified operating conditions. These findings furnish crucial insights into the performance differentials among the matrix-salt adsorbents in the context of cooling applications.

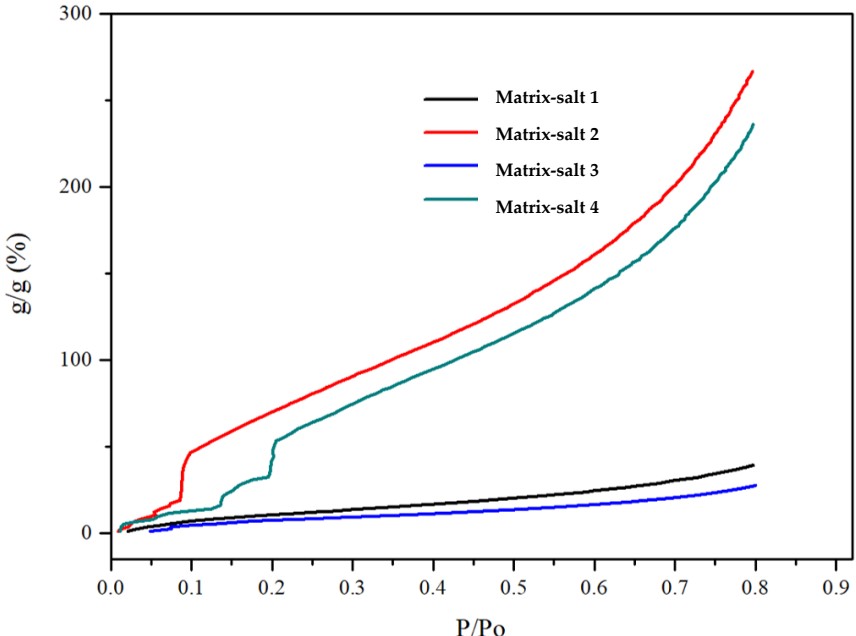

**Figure 5.** Adsorption performance of matrix-salt adsorbents (adsorption temperature: 25 °C).

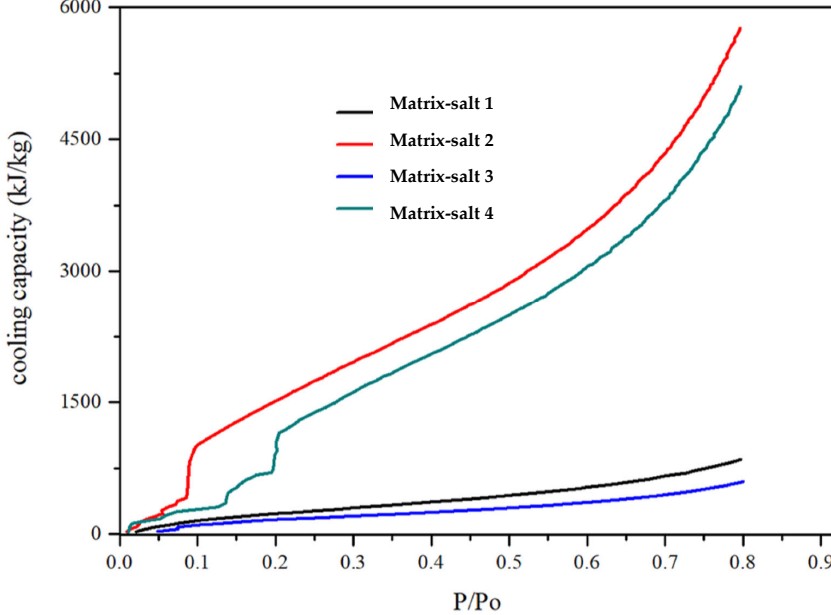

**Figure 6.** Cooling performance of matrix-salt adsorbents (adsorption temperature: 25 °C).

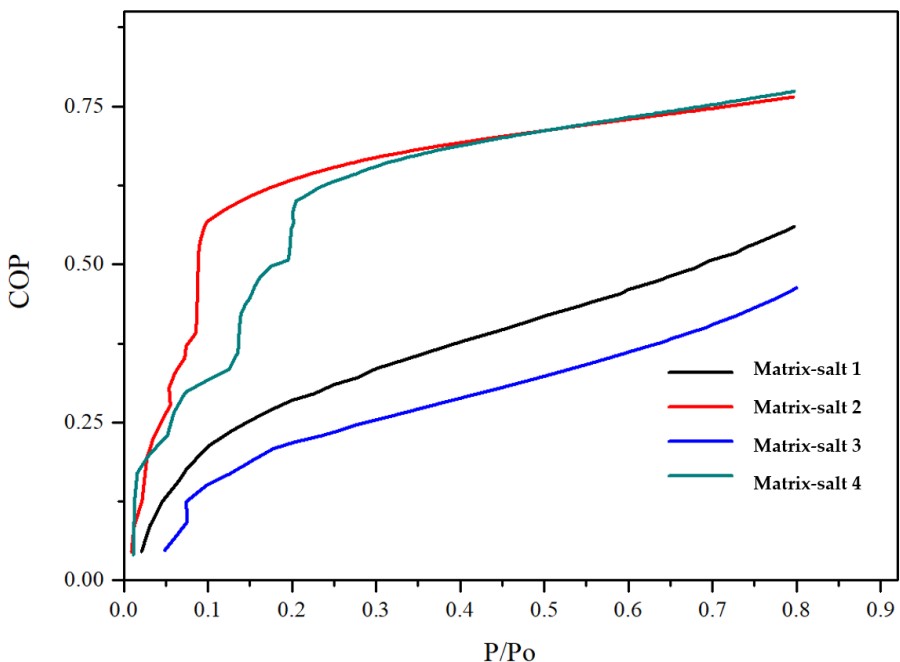

**Figure 7.** Coefficient of performance (COP) of matrix-salt adsorbents (adsorption temperature: 25 °C).

### 3.2. Performance of Heating

Figure 8 provides a visual representation of the heating performance of the matrix-salt adsorbents, while Figure 9 presents data on the heating performance coefficient (COPh). A careful analysis of these graphs reveals that matrix-salt adsorbents featuring vermiculite as the consistent porous substrate showcase commendable adsorption performance. It is worth highlighting that vermiculite-based matrix-salt adsorbents treated with a LiCl solution exhibit a slightly superior adsorption performance when compared to those treated with a CaCl$_2$ solution. This observation is particularly noteworthy at lower p/po (relative pressure) values, where matrix-salt 2 outperforms matrix-salt 4 in terms of adsorption performance. These findings collectively suggest that vermiculite-based matrix-salt adsorbents, especially those impregnated with LiCl, hold significant promise for efficient heating applications. Matrix-salt 2, in particular, stands out as a potential candidate for such applications, especially under lower-pressure conditions. These insights shed light on the potential of these matrix-salt adsorbents in the realm of heating performance and underscore the advantages of utilizing vermiculite and LiCl as key components in their formulation.

The graphical representations also unveil a notable trend in which the adsorption heating performance exhibits enhancement as the evaporator pressures increase. For matrix-salt adsorbents maintaining vermiculite as the consistent porous substrate, there emerges a tendency for the heating performance coefficient (COPh) to converge as the relative pressure (p/po) ascends. Of remarkable significance is the prominent position of matrix-salt 2, which distinguishes itself with the most efficient heating performance among the parameters considered. Conversely, matrix-salt 3 exhibits the least effective heating performance within the specified parameters. To provide a concrete illustration, at an adsorption bed temperature of 25 °C and a relative pressure (p/po) of 0.8, matrix-salt 2 achieves an impressive heating performance of 9920.8 kJ/kg, coupled with a COPh of 1.51. This is because during the adsorption heating process, both adsorption heat and condensation heat can be utilized, resulting in a COPh greater than 1. In stark contrast, under identical conditions, matrix-salt 3 manages to attain only a modest heating performance of 1582.7 kJ/kg, accompanied by a significantly lower COPh of 1.22. These findings underscore the noteworthy potential of matrix-salt 2 for efficient heating applications, particularly under conditions of elevated pressure. The observed trends further emphasize the advantages of employing vermiculite-based matrix-salt adsorbents in such heating

scenarios, with matrix-salt 2 emerging as a particularly promising candidate for optimizing heating performance.

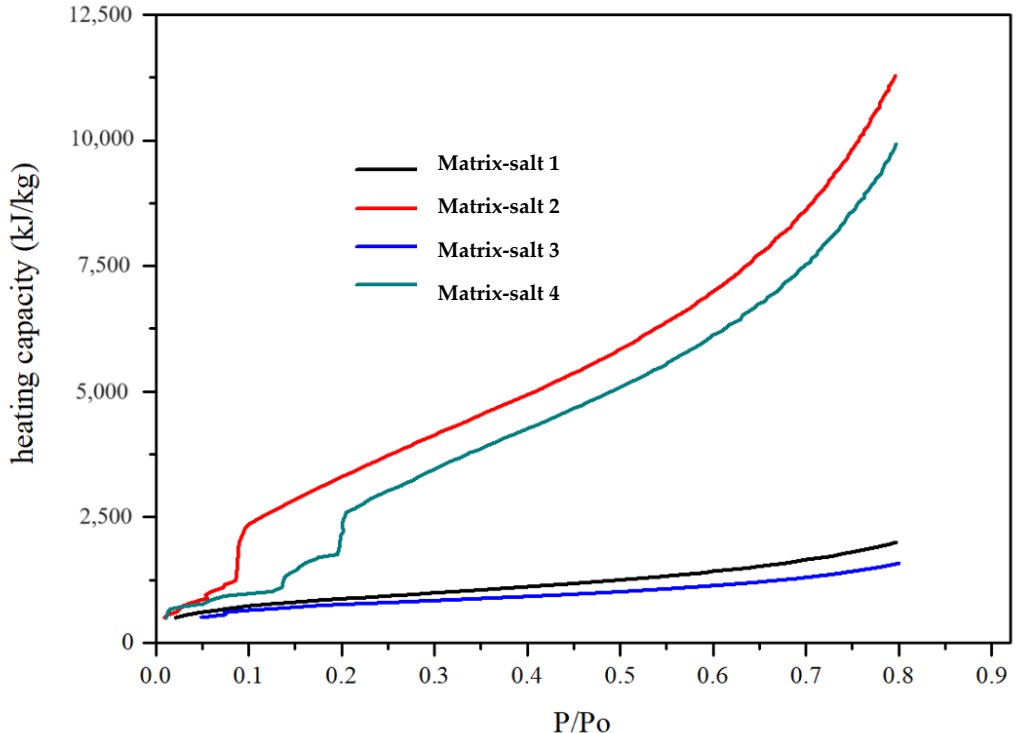

**Figure 8.** Heating performance of matrix-salt adsorbents (adsorption temperature: 25 °C).

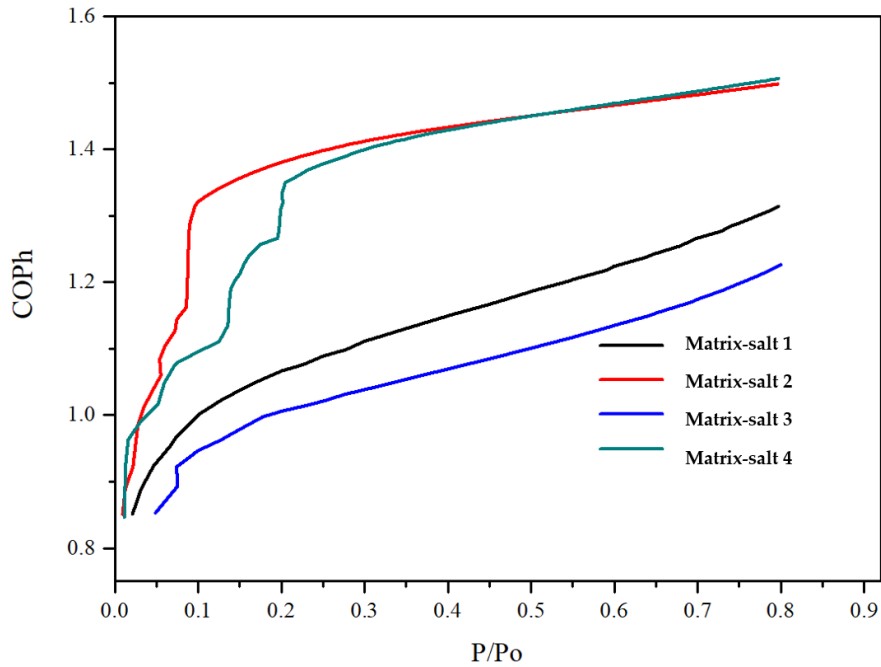

**Figure 9.** Coefficient of performance for heating (COPh) of matrix-salt adsorbents (adsorption temperature: 25 °C).

### 3.3. Performance of Power Generation

The process of desorption in matrix-salt adsorbents generates high-temperature and high-pressure steam, which can be harnessed as a valuable resource to drive a turbine generator, thereby contributing to electricity generation. In Figure 10, we are presented

with an overview of the performance of electricity generation within this context. A detailed analysis of the graph reveals that the matrix-salt adsorption system, leveraging the pressure differential between the desorption bed and the condenser, exhibits a noteworthy capacity for electricity generation. For instance, consider a scenario where the desorption temperature is maintained at 95 °C and the condenser temperature is around 25 °C, with the maximum partial pressure ratio (p/po) set at 0.8. Under these conditions, matrix-salt 1 demonstrates a commendable electricity generation capacity, reaching 10.6 kJ/kg. This is because the high-temperature, high-pressure water vapor desorbed from the adsorption system can drive the operation of a steam turbine, which, in turn, can drive a generator to produce electricity. The comparison between other studies and the results of this paper is shown in Table 3. Upon comparison, it was found that the vermiculite/$CaCl_2$/LiCl system employed in this paper exhibited superior adsorption performance, refrigeration capacity, and COP (coefficient of performance). Its adsorption performance was 32.5% higher than that of the activated carbon fiber/$CaCl_2$/LiCl system, and its COP was 55.1% higher than that of the zeolite 13X/$CaCl_2$ system.

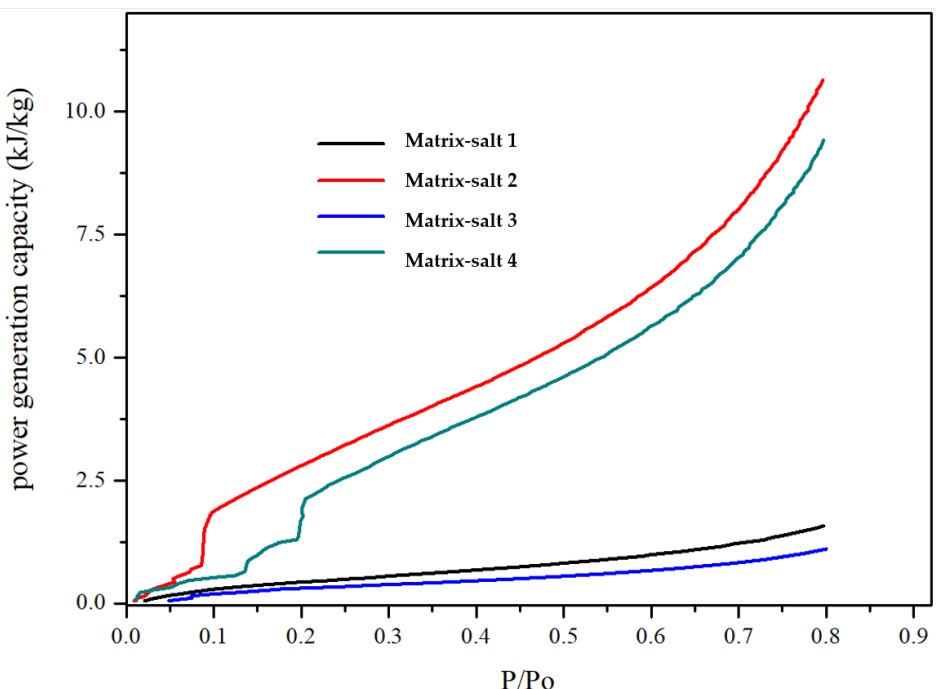

**Figure 10.** Expansion work and power generation performance of matrix-salt adsorbents.

**Table 3.** Comparison between other studies and the results of this paper.

| Type | Adsorbent | Salt Solution Components | | | | |
|---|---|---|---|---|---|---|
| | | Adsorption g/g% | Cooling kJ/kg | Heating kJ/kg | COP | COPh |
| Matrix-salt 1 | Activated carbon/$CaCl_2$/LiCl | 39.3 | 850.6 | 1996.1 | 0.55 | 1.31 |
| Matrix-salt 2 | Vermiculite/$CaCl_2$/LiCl | 266.7 | 5760.7 | 11,286.5 | 0.76 | 1.49 |
| Matrix-salt 3 | AC/Diatomite/$CaCl_2$/LiCl | 27.6 | 597.1 | 1582.7 | 0.46 | 1.22 |
| Matrix-salt 4 | Vermiculite/$CaCl_2$ | 235.9 | 5096.4 | 9920.8 | 0.77 | 1.50 |
| Reference [21] | Zeolite 13X/$CaCl_2$ | / | 523.4 | / | 0.49 | / |
| Reference [22] | FAPO4-5/$CaCl_2$ | 136.6 | / | / | / | / |
| Reference [23] | LiCl + $CH_3COONa$)/ACF/$SiO_2$ | 149.7 | / | / | / | / |
| Reference [24] | Activated carbon fiber/$CaCl_2$/LiCl | 201.3 | 1476.2 | / | / | / |

## 4. Discussion and Conclusions

These results demonstrate that different matrix-salt adsorbents exhibit varying adsorption performance, with the zeolite-CaCl$_2$/LiCl adsorbent showing the highest adsorption performance while maintaining cost-effectiveness. Furthermore, different matrix-salt adsorption systems possess diverse refrigeration capabilities. Among them, the zeolite-CaCl$_2$/LiCl adsorption system demonstrates the highest refrigeration performance and coefficient of performance (COP). This adsorption refrigeration system can be driven by solar energy as well as industrial waste heat. Various matrix-salt adsorption systems also exhibit distinct heating capabilities. Among them, the zeolite-CaCl$_2$/LiCl adsorption system exhibits the highest heating performance and coefficient of performance for heating (COPh). This adsorption heating system can be utilized during the winter season. Additionally, the zeolite-CaCl$_2$/LiCl adsorption system can be employed for both desalination of seawater and electricity generation. It is evident that the zeolite-CaCl$_2$/LiCl adsorption system can fulfill multiple functions, making it a practical, cost-effective, and environmentally friendly solution.

This study represents a comprehensive exploration of the development and characterization of four distinct matrix-salt adsorbents, employing advanced techniques and rigorous measurements. Confocal microscopy was employed to unveil detailed micro-scale insights into the structural attributes of these matrix-salt adsorbents. Additionally, an adsorption analyzer was utilized to conduct exhaustive tests, meticulously assessing the adsorption performance of these multifaceted materials. Subsequently, a systematic analysis was undertaken to evaluate the cooling, heating, and electricity generation capabilities of these matrix-salt adsorbents, yielding valuable insights into their potential applications. To begin, the vermiculite-based matrix-salt adsorbents were found to possess a distinctive morphology, characterized by a "worm-like" structure, layered architecture, and a notably substantial pore volume. Moreover, it was evident that vermiculite matrix-salt adsorbents treated with a LiCl solution outperformed their counterparts treated with a CaCl$_2$ solution in terms of adsorption performance. However, the use of calcium chloride can effectively enhance the economic viability of the adsorption system, given that the price of calcium chloride is approximately one-thirteenth that of lithium chloride. This performance superiority was particularly pronounced at lower p/po (partial pressure ratio) values, underscoring the favorable adsorption characteristics of LiCl-impregnated vermiculite matrix-salt adsorbents. Furthermore, under specific operational conditions featuring an adsorption bed temperature of 25 °C and a p/po partial pressure of 0.8, matrix-salt 2 exhibited remarkable adsorption cooling performance, achieving a cooling capacity of 5760.7 kJ/kg with a coefficient of performance (COP) of 0.75. Additionally, it showcased an impressive heating capacity of 9920.8 kJ/kg with a heating COPh of 1.51, along with a noteworthy electricity generation capacity of 10.6 kJ/kg. These collective findings underscore the multifaceted and promising applications of these matrix-salt adsorbents across various energy domains, including cooling, heating, and electricity generation. This positions them as compelling candidates for the development of sustainable and efficient energy systems, showcasing their potential to contribute significantly to the advancement of clean and renewable energy technologies.

**Funding:** This work was sponsored by the National Natural Science Foundation Project (52271323), Natural Science Foundation of Chongqing, China (cstc2021jcyj-msxmX1092), the Ministry of Education Industry-University Cooperation Collaborative Education (220606517274858, 202102168008, 202102464053), key projects in teaching research and practice of energy and power in higher education institutions (NDJZW2021Z-21), Shanghai Jiao Tong University Decision Consulting (JCZXSJA2022-05), Shanghai Jiao Tong University innovation and Entrepreneurship Special Fund, responsible person, construction of innovation and entrepreneurship talent training system, (CTLD23C 0006 & CTLD23J0033), Shanghai Jiao Tong University Overseas Student Research Practice Base, Projects for Higher Education Scientific Research Planning (23SYS0104), PRP Projects, and software of Simdroid.

**Institutional Review Board Statement:** Not applicable.

**Informed Consent Statement:** Not applicable.

**Data Availability Statement:** The raw/processed data required to reproduce the above findings cannot be shared at this time as the data also forms part of an ongoing study.

**Acknowledgments:** We Thank W.L. Luo, K. Xia, A.F. Cai, H.Y. Shao, W.S. Chen, Y.H. Wang, and A. Rehman for the test and experiment setup.

**Conflicts of Interest:** The author declares that he has no known competing financial interests or personal relationships that could have appeared to influence the work reported in this paper.

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
