# Peer review of "Comprehensive Investigation of Cooling, Heating, and Power Generation Performance in Adsorption Systems Using Compound Adsorbents: Experimental and Computational Analysis"

_sustainability, doi:10.3390/su152115202_

Round 1

Reviewer 1 Report

The article does not define what has been followed as a method comprehensively. It may be improved by the data produced for similar systems previously detected. A reference material giving similar results to the literature may be useful to prove the correctness of the material testing. The findings are all remarkable but should be checked for correctness with an analysis done before for a known sample.

The last paragraph of section 1 should be clarified and supported by some literature to prove novelty.

Fig 1 is scant. It should be improved.

The material systems may be defined with better wordings instead of compound. The wording must be more descriptive to follow more easily in the text.

Figure 2 may be completed with a block diagram. Figure 3 is meaningless and can be discarded.

Units must follow the words after a space.

Discussion about the results is poor.

The manuscript seems not to be prepared carefully enough in writing. Considerable bad wording and grammatical errors are there in the text. They should be corrected before publication.

Author Response

We thank the reviewers for their careful read and thoughtful comments on the previous draft. The manuscript has been amended and the revised version is returned to you now. A detailed response to each of the reviewers` comments is together with the revised paper. Please reconsider the manuscript for possible publication.

Reviewer 2 Report

This article investigated the performance of cooling, heating, and power co-generation systems based on experimental and computational analysis. Following are the comments to the authors.

1. The product model and specification of the equipment used in the experiment should be given.

2. The manuscript in section 3 should be aligned left and right. 3. The results displayed in section 3 should be described with much more physical reasons, not just a tendency or quantitative description. 4. Section 4 and Section 5 can be combined.

No comments.

Author Response

(The authors gave the same response as above.)

Reviewer 3 Report

This paper aims to contribute to sustainable development. As shown in the research, the results obtained that  adsorption performance of vermiculite compound adsorbents impregnated with LiCl solution is superior to those impregnated with CaCl2 solution, with the former exhibiting adsorption at lower p/po partial pressure ratios.

1) Introduction

I suggest to add more literature. As far as I'm concerned, 18 references for an introduction section is not enough.

The aim of the paper seems to be described well in the introduction section.

2. Compound adsorbents and test set-up section

I suggest to add a method overview or a brief introduction of the method rather than entering in detail so early.

I found the rest of the section well explained and justified.

3. Results of performance of cooling, heating, and power generation

3.1. Performance of cooling

I find the section well written. However, I would add more deeper discussion in order to compare your results with current literature.

4. Discussion section

This section must be improved. There is almost nothing. I would like to know different assessment of your findings such as the economics aspects.

I find the paper interesting and well written but it must be improved. Clearly the literature review is poor (only 18 references for a scientific paper).  Due to this and also the discussion section and other topics, I think that the paper needs to be reworked.

Author Response

(The authors gave the same response as above.)

Round 2

Reviewer 1 Report

The manuscript is well written. The tests performed are all relevant to the study.

The numbers are given in the correct shape and order.

Figures and tables are all clear.

Also, it seems it has been corrected before and changed according to the previous reviewers' recommendations.

Discussions and conclusions are all relevant to the ext and comprehensive.

As I see, there are only some mistakes to be corrected in the references part. All were not consistent in writing style.

Reviewer 3 Report

Dear Authors,

I find that the paper has been improved based on reviewers' comments.

I recommend it for publication.